# Copper and Copper/Zinc Ratio in a Series of Cystic Fibrosis Patients

**DOI:** 10.3390/nu12113344

**Published:** 2020-10-30

**Authors:** Marlene Fabiola Escobedo-Monge, Enrique Barrado, Carmen Alonso Vicente, María Antonieta Escobedo-Monge, María Carmen Torres-Hinojal, José Manuel Marugán-Miguelsanz, María Paz Redondo del Río

**Affiliations:** 1Faculty of Medicine, Valladolid University, Avenida Ramón y Cajal, 7, 47005 Valladolid, Spain; mctorresh@telefonica.net (M.C.T.-H.); paz.redondo@uva.es (M.P.R.d.R.); 2Department of Analytical Chemistry, Science Faculty, Campus Miguel Delibes, University of Valladolid, Calle Paseo de Belén, 7, 47011 Valladolid, Spain; ebarrado@qa.uva.es; 3Department of Pediatrics of the Faculty of Medicine, Valladolid University, Section of Gastroenterology and Pediatric Nutrition, University Clinical Hospital of Valladolid, Avenida Ramón y Cajal, 7, 47005 Valladolid, Spain; carmenalonso@gmail.com (C.A.V.); jmmarugan@telefonica.net (J.M.M.-M.); 4Department of Chemistry, Science Faculty, University of Burgos, Plaza Misael Bañuelos sn, 09001 Burgos, Spain; antoitalia777@gmail.com

**Keywords:** hypocupremia, hypercupremia, inflammatory response, risk of zinc deficiency, serum copper/zinc ratio

## Abstract

Cystic fibrosis (CF) patients require a stable and sufficient supply of micronutrients. Since copper is an essential micronutrient for human development, a cross-sectional study was carried out to investigate the serum copper levels, serum copper/zinc (Cu/Zn) ratios, and their relationship with nutritional indicators in a group of CF patients. Anthropometric, biochemical, and dietary measurements, an abdominal ultrasound, and respiratory and pancreatic tests were conducted. Seventeen CF patients were studied (10 females, 59%), 76.5% of whom were ∆F580. Their mean serum copper (113 ± 23 μg/dL) was normal, and there was only one teenager with hypocupremia (6%) and two children with hypercupremia (18%). A significant association between serum copper and zinc levels was discovered. The Cu/Zn ratio was higher than 1.00 for 94% of patients, which is an indicator of an inflammation status. There was no significant correlation between the serum copper concentrations and respiratory and pancreatic function, respiratory colonization, and the results of the abdominal ultrasound. Linear regression analysis showed that serum copper had a positive association with both the Z-score body mass index (BMI) and mean bone conduction speed (BCS). Therefore, since 94% of CF patients had a Cu/Zn ratio > 1.00, this factor must alert us to consider the risk of zinc deficiency and high inflammatory response. The measurement of serum zinc alone does not show one’s zinc status. However, the Cu/Zn ratio may be an indicator of zinc deficiency and the inflammatory status of CF patients.

## 1. Introduction

Cystic fibrosis (CF), also called mucoviscidosis, is recognized as an important genetic disease worldwide [1]. It is an autosomal recessive disorder that commonly affects white people with an annual incidence of approximately 1 in 3500 live births [2]. This multisystem disorder is characterized by genetic mutations in the CF transmembrane conductance regulator (*CFTR*) gene on chromosome 7, which encodes a protein that is essential for the regulation of transmembrane chloride reabsorption [3]. Mutations in *CFTR* result in channelopathy, with impaired sodium and chloride conductance obstructing the mucosa of the exocrine glands [4], affecting a variety of organs, including the lungs, pancreas, intestine, and hepatobiliary tract [5]. More importantly, CF is characterized by a progressive lung infection and exocrine pancreatic dysfunction due to the production of altered sweat and increased mucus production in the lungs and digestive system [6]. Lung disease is the most closely associated cause of morbidity and mortality in these patients [7]. However, pancreatic disease presents the highest penetrance regarding the severity and consequences of CF, as the pancreas is one of the first organs to be affected by this disease [8]. 

In CF, there is a strong association between nutritional status and lung function, and therefore life expectancy [9]. With increasing longevity, the burden and prevalence of comorbidities increase, which includes CF-related diabetes (CFRD), CF-related liver disease (CFLD), and CF-related kidney and bone disease, along with the increased chance for obesity and overweight, which were all reported in CF patients [5,10,11]. CF is closely related to a poor nutritional status, which is linked directly to factors associated with the genetic mutation underlying this disease [12]. In addition to a decreased nutrient intake, especially during periods of acute illness [13], the risk of nutritional deficiencies in CF patients is likely to be due to several coexisting factors, such as the malabsorption of fat, protein, energy, and micronutrients that are secondary to pancreatic insufficiency, the alteration in bile salts, the increase in energy needs due to the deterioration in lung function, chronic inflammation, and not only microbial colonization but also recurrent lung infection [14]. Although the prevalence of fat-soluble vitamin deficiencies (vitamins A, D, E, and K) and the need for their supplementation in CF is well known [13], other micronutrient deficiencies, such as minerals and trace elements, are not well established, especially during acute exacerbations [15].

Copper is an enigmatic ion that has an important role in biological systems [16]. It is a transition metal that exists in two forms: Cu^+^ is the reduced cuprous form and Cu^2+^ is the oxidized cupric form of copper [17,18]. This important dietary ion fulfills essential structural functions in enzymes [16], such as cytochrome oxidase, superoxide dismutase, monoamine oxidase, and lysyl oxidase [19]. Furthermore, it is considered a critical cofactor for a group of cellular transporters, namely, the cuproenzymes [17]. Organ meats, nuts, seeds, chocolate, and shellfish are rich sources of copper [18]. However, due to the ability of copper to alternate between two oxidation states, and since free copper is toxic, most of the cellular copper is tightly bound [16]. Copper is essential for the proper functioning of the human body, as it mainly intervenes in metabolic processes, such as the synthesis of hemoglobin, the function of neurotransmitters, the oxidation of iron, cellular respiration, amidation of antioxidant peptides, and the formation of pigments and connective tissue [17]. Copper is necessary for growth, defense mechanisms, bone mineralization, the maturation of red and white blood cells, iron transport, cholesterol metabolism, myocardial contractility, glucose metabolism, and brain development [20]. 

Even though its deficiency is more commonly an acquired condition that is induced by the imbalance between need and dietary copper supply [21], it has been reported in subjects with malabsorption of copper due to malabsorption syndromes, such as celiac disease, tropical and nontropical sprue, CF, and short-bowel syndrome [22]; resulting from intestinal resection (gastric surgery, including gastric bypass or gastrectomy) [23]; the excessive use of copper chelators, antacids, zinc supplement overuse, parenteral overdosing, and denture cream ingestion with zinc; chronic parenteral nutrition without proper copper supplementation and prolonged jejunal enteral feeding; a diet low in copper; other unknown causes [24]. Furthermore, copper deficiency can occur in premature infants who are fed formulas with inadequate copper content, newborns with chronic diarrhea or malnutrition, and patients undergoing prolonged dialysis or who have suffered severe burns [25]. Since copper is involved in many bodily functions, its deficiency can cause a wide range of symptoms [26] that occur in stages of increasing severity (marginal, moderate, and severe clinical deficiencies) [27]. Anemia, neutropenia, and bone abnormalities are the most frequent clinical manifestations of copper deficiency [21]. 

Nevertheless, although serum copper is a reliable indicator of its deficiency and falls to very low concentrations in people with copper deficiency, it does not reflect dietary intake, except when the intake is below a certain level [28]. In addition, the serum concentration of copper returns to normal values a few days after its supplementation [20]. The ineffective absorption of copper from the diet, or excessive loss of copper through bile, can cause systematic copper deficiency and it occurs in CF [25]. Furthermore, as is well known, the nutritional status of CF patients has a great impact on their life expectancy, and continuous monitoring to improve their nutritional status is a primary goal of treatment in each patient [29]. For this reason, it was hypothesized whether an abnormal serum copper level was prevalent in a series of CF patients under nutritional control. Therefore, the main objective of this study was to investigate the serum copper levels, the copper/zinc ratio, and its relationship with nutritional indicators in a group of patients with cystic fibrosis. The nutritional status of zinc in this group has already been published [30].

## 2. Materials and Methods

The study design was cross-sectional and was conducted at the Nutrition Unit of the Pediatrics Service at the University Clinical Hospital in Valladolid for 18 months. The study was conducted according to the Declaration of Helsinki and the protocol was approved by the University Clinical Hospital Ethics Committee (INSALUD-Valladolid, 14 February 2002). The study population included both pediatric and adult CF patients. All subjects gave their informed consent for inclusion before they participated in the study. The inclusion criteria were children with a proven diagnosis of CF. Exclusion criteria were refusal to take part, hospital admission, and suffering from an acute infection. 

Detailed study methods regarding assessing zinc nutritional status in this series are reported elsewhere. Briefly, this nutritional study included an assessment of phenotypic characteristics, a blood test, a dietary survey, and a specific assessment of respiratory and pancreatic functions [30]. Figure 1 shows the flowchart of this cross-sectional study. In this series, the genotype of the participants, as well as the Norman–Crispin score (>5), the forced vital capacity (FVC% < 80%) and the forced expired volume in 1 s (FEV1 < 80%) to estimate respiratory sufficiency (SR) and insufficiency (IR), and the fat absorption coefficient (CFA > 94%) to evaluate pancreatic sufficiency (PS) and insufficiency (PI) were studied and published [30]. An abdominal ultrasound was taken to assess the digestive tract status. 

In addition to the evaluation of weight (W) and height (H), BMI, BMI–height–age, and their corresponding indicators, Z-score, and growth rate were measured [30,31,32]. The circumference of the wrist, hip, waist, mid-arm, triceps, biceps, and subscapular and suprailiac folds (Holtain skinfold gauge, pressure 10 g/mm^3^) were measured to obtain the waist–hip ratio, waist–height index, muscular area of the middle of the arm (MAMA), fat-free mass (FFM), and fat mass (FM). Body composition was measured using anthropometry and bioelectrical impedance analysis with an RJL Systems single frequency impedance analyzer [RJL BIA-101 (RJL System, Detroit, MI, USA)] (source). Bone densitometry by ultrasound [DBM Sonic 1200 IGEA (Emsor S.A., Madrid. Spain] was measured through the BCS of the last four fingers of the non-dominant hand [33]. Basal energy expenditure (EE) or resting EE (REE) was measured using fasting indirect calorimetry (IC) with a canopy system [Deltatrac II (Datex-Ohmeda. Helsinki. Finland)] in standardized conditions.

Apart from the evaluation of a complete blood count, the blood analysis included biochemical analysis of acute-phase proteins, C reactive protein (CRP), and the erythrocyte sedimentation rate (ESR); the evaluation of the serum concentration of copper and zinc [30] was done using atomic absorption spectrophotometry (model PU9400 Philips Scientific, Cambridge, UK) [34]. Less than 70 µg/dL and more than 140 µg/dL were the cut-offs used to categorize hypo- and hypercupremia, respectively [35]. The Cu/Zn ratio as an alternative biomarker for assessing inflammatory and nutritional status and adverse clinical outcomes was measured [36], where its normal values range from 0.7 to 1.0 [37]. The zinc/copper (Zn/Cu) ratio was also evaluated.

A 24 h food diary was recorded for 3 days and the collection of information on the habitual diet of the participants was verified using an in-depth interview (approximately 90 min). Through this prospective dietary survey of 72 h, including a weekend day, the daily energy intake (IE); proteins; carbohydrates; lipids; monounsaturated, polyunsaturated, and saturated fats; fiber; vitamins A, B1, B2, B6, B12, C, D, and E; niacin; folic acid; calcium; magnesium; iron; iodine; zinc intake were measured using the Mataix Food and Health computer program [38]. Less than 80% of the percentage of the Dietary Reference Intake (% DRI) and more than 120% DRI were considered inadequate intakes. Besides pancreatic enzyme replacement therapy (PERT) and fat-soluble vitamin supplements, the patients did not receive zinc or copper supplements. 

Data distributions are described as mean and SD or median (including lower and upper quartiles). Comorbidities in this series are expressed as percentages. A comparison between groups for continuous and categorical variables was performed using the Mann–Whitney *U* and McNemar tests, respectively. Spearman’s correlation was performed to test the associations. The analysis of variance (Kruskal–Wallis test) was used to search for interactions according to gender and copper level. Simple and multiple linear regression analysis was calculated to study the relationships between two or more correlations. The analyses were performed using IBM SPSS version 24.0 (IBM Corp., Armonk, NY, USA). The level of significance was set at *p* < 0.05.

## 3. Results

The baseline demographics and clinical characteristics for the 17 participants (10 females, 59%) are reported elsewhere (Table 1) [30]. The mean age was 14.8 ± 8 years with a median of 15 and a range of 2–31 years (seven children, five adolescents, and five adults). Briefly, the most frequent mutation was the homozygous Delta F580 (41%). No patients had stunted growth but 29.4% (5/17 cases) displayed undernutrition. There was no obesity according to the BMI but there were two overweight patients (8- and 13-year-olds) and obesity (2- and 25-year-olds) according to the waist-to-height index. Although there were no differences in the anthropometric measures according to sex and pancreatic and pulmonary function, a lower BMI was observed in those with PS compared to PI (*p* = 0.020). However, lung function was not significantly worse in the 17.6% of patients colonized by *Pseudomonas aeruginosa* and *Candida sp*. And the 23.5% colonized by *Staphylococcus aureus* compared to those without such colonization. There was a significant difference between the 64.7% of CF patients with PI and a normal abdominal ultrasound result and the 23.5% with PS and an abnormal abdominal ultrasound result (*p* = 0.0001).

The median and mean serum copper of 113 ± 23 μg/dL (Q1–3, 100–125 μg/dL) was normal and it ranged between 69 and 158 μg/dL. Although females patients presented higher serum copper (116 μg/dL) than males (109 μg/dL) and the mean serum copper of children (125 μg/dL) was higher than that of adults (114 μg/dL) and adolescents (96 μg/dL), these differences were not significant. There was no significant difference in serum copper when comparing patients with high and low abdominal ultrasound results, sufficient and insufficient respiratory and pancreatic function, as well as patients with or without respiratory colonization (Table 2). Mean serum copper was significantly lower in the undernourished (90 ± 14.1 μg/dL) than in the eutrophic (122.7 ± 19.7 μg/dL, *p* = 0.004) CF patients. There was a male teenager (6%) with hypocupremia and two female children (12%) with hypercupremia, where all of them were eutrophic. Table 3 shows the significant correlations and regression analysis between serum copper and the nutritional parameters studied. Serum copper did not correlate with dietary zinc, calcium, magnesium, and iron intakes. CF patients with RI had more dietary zinc intake (112.5 ± 23.6% DRI) than RS (81.5 ± 20.8% DRI, *p* = 0.015). Serum copper decreased slightly with increasing age (Figure 2), and the serum copper and zinc levels had a direct association when they were adjusted by age (Figure 3). 

The mean Cu/Zn ratio of 1.32 ± 0.28 (Q1–3, 1.19–1.38) and range from 0.73 to 2.0 was higher than the normal value (range from 0.7 to 1.0) [37] and 94% of CF patients had a Cu/Zn ratio > 1.00. The mean Zn/Cu ratio was 0.79 ± 0.19 (Q1–3, 0.72–0.84) and ranged from 0.5 to 1.38. The Cu/Zn ratio had a direct and significant correlation with protein, monosaturated lipids, niacin intake, triglycerides, and gamma-glutamyl transpeptidase, and a negative association with polyunsaturated lipids intake and monocytes. Between the Cu/Zn and Zn/Cu ratios, there was a negative association (*r* = −0.998; *p* = 0.000). The found Zn/Cu ratio had almost the same associations but in an inverse relationship as the Cu/Zn ratio. All of them had a low percentage of lymphocytes T (CD3) and Th (CD4). Although the mean complement C3 (117 mg/dL, Figure 4), complement C4 (21 mg/dL, Figure 5), and lymphocytes NK CD16+56 (10.8%) were normal, linear regression analysis showed that all of them had a positive association with serum copper. CRP was slightly increased in the adolescent with hypercupremia. The CRP and ESR levels were correlated with the Cu/Zn ratio.

The mean basal EE was lower than the theoretical basal EE (*p* = 0.001) but was adequate according to the World Health Organization’s (WHO’s) recommendation (*p* = 0.074). The mean diet was high in protein with adequate carbs, fiber, and EI. The diet was adequate, except for the low iodine intake. Serum copper did not correlate with the zinc, calcium, magnesium, and iron intakes. CF patients with RI had more zinc intake (112.5% DRI) than RS (81.5% DRI, *p* = 0.015). Only vitamin C intake had a negative association with serum copper (Table 3).

## 4. Discussion

Cystic fibrosis is a multisystem disorder involving the pulmonary, gastrointestinal, endocrine, musculoskeletal, and the male genitourinary systems, as well as the sinuses [4]. PI is one of the main factors of CF morbidity. More than 85% of CF patients show evidence of malabsorption from exocrine PI [39], leading to fat malabsorption, predisposing them to a severe deficiency of fat-soluble vitamins (A, D, E, and K) and trace elements, such as calcium, magnesium, iron, copper, and zinc [40]. Although the information on serum copper levels in patients with CF is scanty [4], the mucus (sputum) of patients with CF reveals that there are traces of metals, mostly iron and copper, but also zinc [6], and in separate in vitro studies, these metals have been shown to induce *Pseudomonas aeruginosa* resistance to carbapenem antibiotics [41]. Additionally, deficiencies of copper can result in iron deficiency anemia [42], osteoporosis and joint problems [43], and increased susceptibility to infection that is secondary to poor immune function [44]. Therefore, the main aim of this study was to investigate the serum copper level, Cu/Zn ratio, and its relationship with nutritional indicators in CF patients. 

Even though there were few CF patients in this study, we must consider that both the median and mean serum copper (113 ± 23 μg/dL) were normal (range from 70 to 140 μg/dL) [35] and agreed with the mean serum copper (115 ± 91 μg/dL) in twenty-seven north Indian children with CF (from 3 months to 12 years) [45]. Nevertheless, it was significantly lower in comparison to the mean serum copper (134.5 ± 25.7 μg/dL, *p* = 0.004) found in healthy children aged 1 to 18 years who had not received any vitamin or mineral supplements [46]. Although low levels of serum and plasma copper, ceruloplasmin, and superoxide dismutase in red blood cells can show a severe copper deficiency, they are not sensitive to a marginal copper state [47]. Furthermore, the response to inflammation and infection can alter serum copper and not determine its deficiency [48]. According to different studies, pediatric reference intervals for serum copper are often difficult to establish [49]. Moreover, patients with CF serum copper and ceruloplasmin levels show variable results [50]. We must bear in mind that the consequences of borderline copper deficiency may have little effect on a normal individual but may have more serious consequences for CF patients [48]. Therefore, its assessment is essential for CF patients.

Milne et al. reported that serum copper changes according to age and sex [51]. Regarding sex, although females had a higher serum copper (116 μg/dL) than males (109 μg/dL), this difference was not significant. Similarly, other studies showed that women had significantly higher serum copper levels (*p* < 0.05) than men [52]. Although mean serum copper was higher in children (125 μg/dL) than adults (114 μg/dL) and adolescents (96 μg/dL), these differences were not significant. However, it was observed that the serum copper decreased slightly with increasing age (Figure 2) in a similar way to those reported by Lin et al. [47]. Moreover, although Best et al. found a moderate copper deficiency in CF patients [48], in this group of patients, there was only one 15-year-old male teenager (6%) with hypocupremia (69 μg/dL). This result contrasts with 44% of CF children with copper deficiency reported by Yadav et al. in twenty-seven north Indian children with CF [45]. According to Cordano, a serum copper concentration of <90 μg/dL and particularly <45 μg/dL lends strong support to the diagnosis of deficiency [53], which is in agreement with our results. In contrast, in this study, there were two girls, one 6-year-old (158 μg/dL) and a 9-year-old with hypercupremia (18%).

As far as nutritional status was concerned, 29% of CF patients displayed malnutrition (four adolescents and one adult), and patients with PI had a higher BMI than patients with PS (*p* = 0.020). Interestingly, mean serum copper was significantly lower in undernourished CF patients (90 μg/dL) than eutrophic patients (122.7 μg/dL, *p* = 0.004). Not one patient had stunted growth, but there were two overweight patients (an 8- and a 13-year-old) and obesity (a 2- and a 25-year-old) according to the waist-to-height index. This fact is interesting because it has been shown in a meta-analysis that a higher level of serum copper could be associated with the risk of obesity in children and adults [54]. Furthermore, in this series, linear regression analysis showed that there was a direct association between serum copper levels and BMI (*R*^2^ = 0.236, *p* = 0.048). This result is not surprising because, in a large-scale sample of 2233 15–65-year-old subjects, a strong positive correlation was found between serum copper and BMI (*R* = 0.85, *p* < 0.001) [55]. 

CF-related bone disease has increased with life expectancy [5,10,11]. Copper is a micronutrient present in almost every cell in the human body. Approximately 50% of the copper content is stored in bones and muscles (approximately 25% in skeletal muscle), 15% in the skin and bone marrow, 8 to 15% in the liver, and 8% in the brain [56]. In addition, it plays an important role in the synthesis of collagen in the bones and connective tissue [57]. According to Turk, a bone mineral density examination in CF patients should be performed at the age of 8–10 y [5]. Surprisingly, the BCS (0.3 ± 0.9) Z-score was normal and no patient with CF had a low BCS. That is, bone densitometry measured using ultrasound was normal and no patient with CF was at risk of osteoporosis [33]. However, linear regression analysis showed that BCS had a positive correlation with serum copper. In contrast, Chase et al. showed that 44% of children with CF, particularly adolescent girls, have bone demineralization [58]. Among the main risk factors for bone loss in CF are poor nutritional status, vitamin D and K deficiencies, calcium, hypogonadism, glucocorticoid use, physical inactivity, *CFTR* dysfunction, and exacerbations of lung infections. To a lesser extent, deficiencies of copper, phosphorus, magnesium, zinc, essential fatty acids and proteins, and an excess of vitamin A may have etiological roles [59]. 

The mean EE was adequate according to WHO’s recommendation (*p* = 0.074). The mean diet was high in protein with adequate carbs, fiber, and EI. The diet was adequate except for the low iodine intake. Serum copper did not correlate with zinc, calcium, magnesium, and iron intakes. Nevertheless, CF patients with RI had more zinc intake (112.5% DRI) than RS (81.5% DRI, *p* = 0.015). Surprisingly, only vitamin C intake had a negative association with serum copper, and all three CF patients with hypo- and hypercupremia had low vitamin C intake (χ^2^ = 0.046). Various dietary factors, such as carbohydrates, iron, zinc, certain amino acids and proteins, molybdenum, and vitamin C, can have adverse effects on the bioavailability of ingested copper [27]. In experimental animals, supplementation with vitamin C can induce a copper deficiency, but it is not clear whether this also occurs in humans [60]. Nevertheless, two studies in healthy men showed that the activity of ceruloplasmin oxidase may be impaired by relatively high doses of supplementary vitamin C [61]. Likewise, vitamin C inhibits copper absorption, binds or chelates copper, and facilitates its removal [62]. 

The usual pathophysiological features of copper deficiency include anemia, leukopenia, and neutropenia [20]. Copper plays a role in the production of hemoglobin, myelin, melanin, and the normal functioning of the thyroid gland [63]. Furthermore, despite normal serum iron levels, copper deficiency affects the production of hemoglobin because copper is required for the use of iron in bone marrow [64]. In this series, it was found that the 6-year-old girl with RI, PS, and mesenteric adenopathy and was colonized by *Pseudomonas aeruginosa* had hypercupremia, iron deficiency, and slightly high CRP; the 9-year-old girl with RS and PI and was colonized by *Hemophilus influenzae* had hypercupremia; the 15-year-old male with PI and RS and was colonized by *Aspergillus fumigatus* had hypocupremia, prealbumin deficiency, and lymphopenia (1610 cell/mm^3^). Table 4 shows another 8-year-old boy with RI and SI and who was colonized by *Candida sp.* Had a high serum copper level on the border of hypercupremia and iron deficiency. All of them had a low percentage of lymphocytes T (CD3) and Th (CD4). In addition, although the mean complement C3 (117 mg/dL, Figure 4), complement C4 (21 mg/dL, Figure 5), and lymphocytes NK CD16+56 (10.8%) were normal, linear regression analysis showed that all of them had a positive association with serum copper. The CRP and ESR levels were normal and no had association with serum copper. These results agreed with the study of Dizdar et al. [36].

Copper may be important for immune system function, where its deficiency is frequently associated with an increased risk of infection and disturbances in copper homeostasis alter immune system function in rodents [25]. Copper may be necessary for the destruction of bacteria by macrophages and copper deficiency can disrupt factors of the cellular and humoral immune system [65]. However, in this series, there were no significant differences in serum copper by bacterial colonization. In contrast, according to Yadav (2014), serum copper was lower (57 μg/dL) in cases with exacerbation of the disease compared to levels in stable cases (*p* = 0.03) [45]. Likewise, in this series, although 17.6% of the patients were colonized by *Pseudomonas aeruginosa* and *Candida spp.*, and 23.5% by *Staphylococcus aureus*, their lung function was no worse than that of those without such colonization [30]. Songchitsomboon et al. observed a significant increase in serum copper levels in patients with infectious diseases [66]. However, Lee et al. reported that serum copper levels increased significantly several months after recovery from an acute pulmonary exacerbation in CF patients [67]. 

### Copper/Zinc Ratio

Copper and zinc deficiencies are common and underdiagnosed health risks [68]. Both micronutrients are required for cellular metabolism and antioxidant defense systems [69]. Acute infections alter metabolism, while deficiencies increase the risks of infection. While acute infections cause an increase in serum copper in the context of an acute phase response [70], they cause a decrease in serum zinc due to its redistribution in the liver and other tissues [68]. Physiological conditions, such as age and sex, as well as malabsorption, inflammatory condition, and genetics, significantly influence the concentrations of both trace elements [71]. In this series, the median serum zinc (86 μg/dL) was in the normal range of 70 to 120 μg/dL [30]. Although 23% of patients had inadequate zinc intake and 17% serum zinc deficiency, none of the patients with deficient intake had hypozincemia. This situation of deficient zinc intake without hypozincemia alerts to a state of a marginal deficiency of around 41%. In addition, serum zinc was associated with BMI and W/H Z-score and zinc intake was associated with EI and weight-for-age Z-score [30]. Nevertheless, no patient with abnormal serum copper had hypozincemia. Only the teenager with hypocupremia had high calcium and iron intake but low magnesium and zinc intake. Although the serum copper and zinc levels did not correlate with each other and there was no association of their serum levels with the age of the patients, linear regression analysis showed that serum copper had a significant association with serum zinc when adjusted according to age (Figure 3). 

One of the most common trace metal imbalances is elevated copper and depressed zinc [71,72]. The optimal plasma or serum ratio between these two elements is 0.70–1.00 [37], and the normal Cu/Zn ratio in children and adults is close to 1:1 [71,72]. In this series, the mean Cu/Zn ratio was high (1.32) with a range from 0.73 to 2.00, and 94% of CF patients had a Cu/Zn ratio > 1.00. The highest Cu/Zn ratio of 2.00 was in the 6-year-old girl with hypercupremia (158 µg/dL) and normal serum zinc (80 µg/dL). A pattern of high copper and low zinc is characteristic of an inflammatory condition [50], and a Cu/Zn ratio greater than 2 means there is a severe bacterial infection [73]. This situation might indicate that an inflammatory status was prevalent in most CF patients in this series. Conversely, the mean Zn/Cu was 0.75 ± 1.9 and ranged from 0.5 to 1.38. A tissue Zn/Cu ratio < 4 is often associated with increased susceptibility to bacterial and viral infections [62]. In addition, the Cu/Zn ratio had a direct and significant correlation with protein, monosaturated lipids, and niacin intake, as well as triglycerides and gamma-glutamyl transpeptidase, and a negative association with polyunsaturated lipids intake and monocytes. In contrast, the Zn/Cu ratio presented almost the same associations but in an inverse relationship as the Cu/Zn ratio.

According to Osredkar et al., the Cu/Zn ratio is clinically more important than the concentration of either trace metal [63]. It has been reported that the Cu/Zn ratio is a good indicator of various diseases [74] and was proved to be a better predictor of disease severity and/or mortality than copper levels [75]. When high levels of copper and low levels of zinc coexist, they can contribute to diseases such as schizophrenia, hypertension, autism, fatigue, muscle and joint pain, headaches, infantile hyperactivity, depression, insomnia, senility, and premenstrual syndrome [71]. The Cu/Zn ratio has also been related to childhood neurological disorders [72] and assaultive individuals [76]. Additionally, the Cu/Zn ratio is an indicator of the nutritional status of zinc in patients [73]. Zinc deficiency should be highly suspected in individuals with high serum Cu/Zn ratios. Previous studies revealed the validity of the Cu/Zn ratio for the severity of nutritional status, inflammation, oxidative stress, immune dysfunction, and infection associated with zinc deficiency [77]. This fact agrees with the results published previously, where 41% of the cases would have an elevated risk of zinc deficiency [30].

At this point, we need to consider four highlights. First, the median serum copper was normal (113 µg/dL) and the prevalence of abnormal serum copper levels was low (6% of CF patients had hypocupremia and 12% had hypercupremia). Second, this study demonstrated that the serum copper level had a significant association with several of the nutritional parameters studied (body mass index and bone conduction speed, vitamin C intake, serum zinc, complements C3 and C4, and lymphocytes NK CD16+56). Third, the mean Cu/Zn ratio was high (1.32) and 94% of CF patients in this series had a high Cu/Zn ratio > 1.00, and only one patient had a high Cu/Zn ratio of 2. These correspond with a high inflammatory response and severe bacterial infection, respectively. Finally, there was a high risk of marginal zinc deficiency (41%), and although no patient with abnormal serum copper had hypozincemia, serum copper significantly correlated with serum zinc. Considering all the highlights, we should indicate that although the assessed biomarkers of inflammation (ESR and CRP) were normal, the high Cu/Zn ratio should alert us to a condition with a high inflammatory response and could reflect the severity of zinc deficiency. 

The results respond to the main objective of this study and indicate the need to continue studying the relationship between the nutritional status of patients with CF and abnormal copper status in order to understand the essential balance between the copper and zinc statuses. A limitation of this study is the small number of participants, while its strengths lie in the determination of serum copper levels, the Cu/Zn ratio, and its relationship with anthropometric, biochemical, and dietary indicators. We suggest the implementation of multicenter trials to improve the knowledge of copper status in these patients and to determine the necessary and appropriate amount of copper supplementation to improve the nutritional status of cystic fibrosis patients when necessary. 

## 5. Conclusions

The mean serum copper was normal and had a direct association with the nutritional status, expressed as body mass index, bone conduction speed, serum zinc, complements C3 and C4, and lymphocytes NK CD 16+56, and a negative association with vitamin C intake. No patients with hypocupremia (6%) and hypercupremia (12%) had hypozincemia. The mean Cu/Zn ratio was high, and 94% of CF patients had a high Cu/Zn ratio. This situation must alert us to the risk of zinc deficiency and a high inflammatory response.

## Figures and Tables

**Figure 1 nutrients-12-03344-f001:**
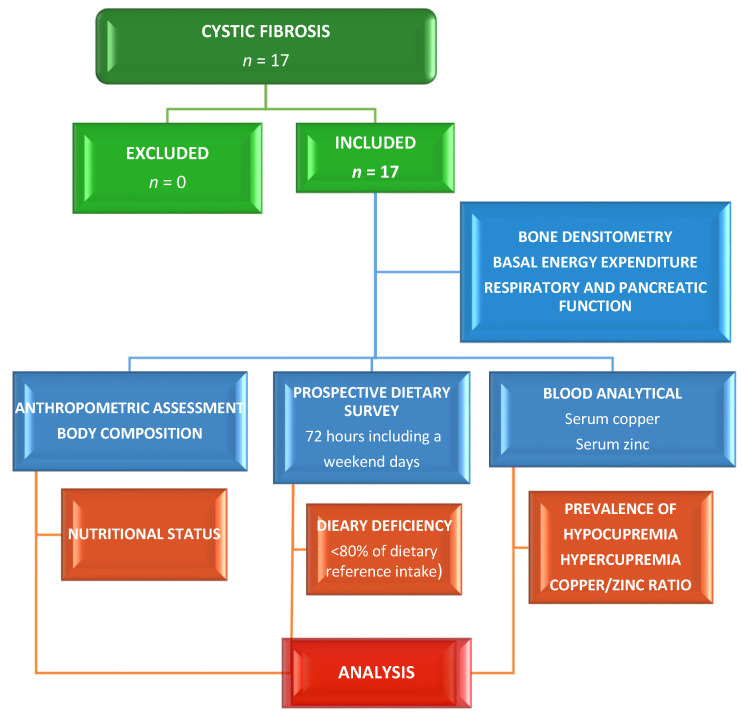
Flowchart of the cross-sectional study.

**Figure 2 nutrients-12-03344-f002:**
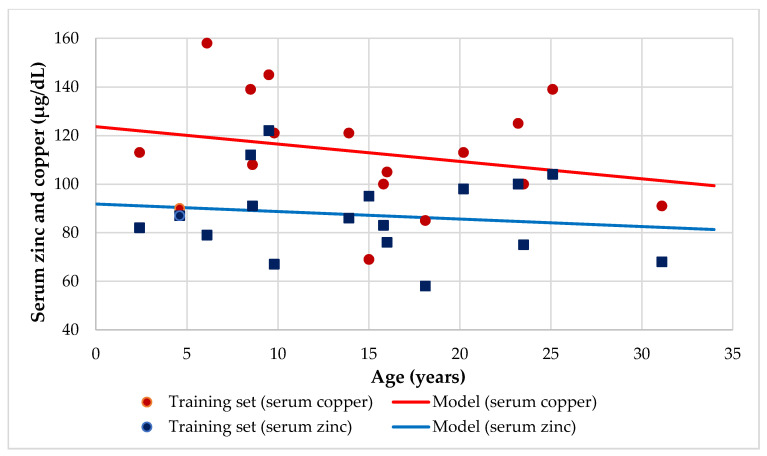
Regression serum copper and zinc (µg/dL) by age.

**Figure 3 nutrients-12-03344-f003:**
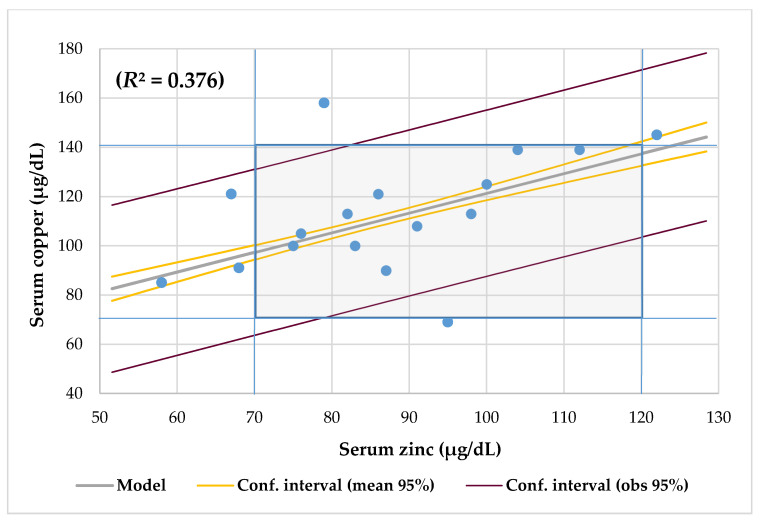
Regression serum copper (70–140 µg/dL) by zinc (70–120 µg/dL) adjusted for age and cut-offs.

**Figure 4 nutrients-12-03344-f004:**
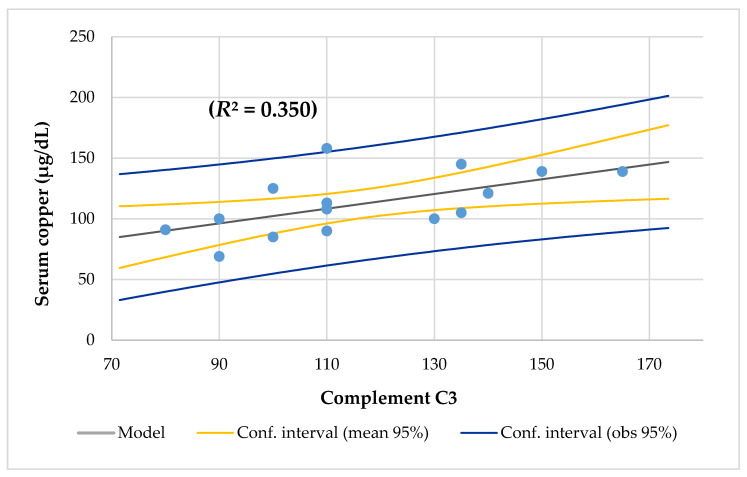
Regression serum copper by complement C3.

**Figure 5 nutrients-12-03344-f005:**
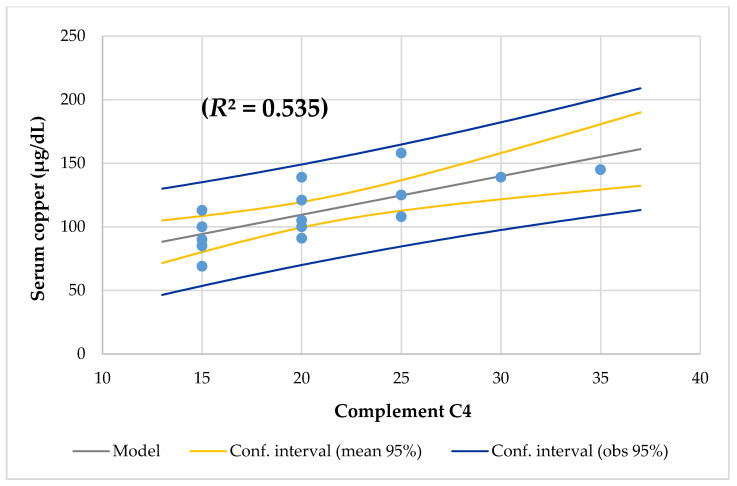
Regression serum copper by complement C4.

**Table 1 nutrients-12-03344-t001:** Baseline demographic and clinical characteristics of participants (*n* = 17 *) [30].

Characteristics	Mean ± SD or No. (%)	Median	Range
Age (years)	14.8 ± 8	15	2–31
Anthropometric Assessment			
Body mass index Z-score	−0.95 ± 1.1	−0.6	−3.8 to 0.6
Average conduction velocity Z-score	0.3 ± 0.9	−0.0	−1.6 to 1.4
Indirect Calorimetry (calories)			
Basal energy expenditure	1078 ± 303	1149	440–1490
Theoretical basal energy expenditure	2193 ± 576	2200	1066–3251
WHO recommended basal energy expenditure	1185 ± 233	1230	598–1559
Blood Analytics			
Serum copper level (µg/dL)	113 ± 23.5	113	69–158
Serum zinc level (µg/dL)	87.2 ± 16.7	86	58–122
Copper/zinc ratio	1.32 ± 0.28	1.33	0.73–2.00
Zinc/copper ratio	0.79 ± 0.18	0.75	0.5–1.38
Complement C3 (mg/dL)	117 ± 24	110	80–165
Complement C4 (mg/dL)	21 ± 6	20	15–35
Lymphocytes NK CD16+56 (%)	10.8 ± 5.8	9	2–22
Gamma glutamyl transpeptidasa (U/L)	20.9 ± 19.3	13	7–70
Prospective Dietary Survey			
Energy intake (calories)	2595 ± 464	2672	1846–3410
Vitamin C intake (percentage of dietary reference intake)	170 ± 141	131	13–461
Zinc intake (percentage of dietary reference intake)	97 ± 26.9	98	54.9–153.9
Comorbidities (%)			
Undernutrition	5 (29.4%)		
Anemia by iron deficiency	5 (29.4%)		
Hypocupremia	1 (5.9%)		
Hypercupremia	5 (29.4%)		
Hypozincemia	3 (17.6%)		
Dietary zinc deficiency	4 (23.5%)		
Abnormal abdominal ultrasound	5 (29.4%)		

* Seventeen cystic fibrosis patients were screened, included, and analyzed. WHO: World Health Organization.

**Table 2 nutrients-12-03344-t002:** Differences between fibrosis cystic patients.

Characteristics	Male	Female	*p*-Value
Age (years)	10.4 ± 7.2	17.2 ± 7.9	0.091
Serum copper level (µg/dL)	109.3 ± 23.4	115.7 ± 24.5	0.596
Colonization	Yes	No	
Serum copper level (µg/dL)	119 ± 24	102.2 ± 19.8	0.098
Forced vital capacity	76.9 ± 24.2	94.5 ± 53.7	0.478
Forced expired volume in 1 second	74.9 ± 27.3	84.9 ± 27.5	0.511
Nutritional Status	Undernutrition	Eutrophic	
Serum copper level (µg/dL)	90 ± 14.1	122.7 ± 19.7	0.004
Acute Phase Reactants	ERS high	Normal	
Serum copper level (µg/dL)	116.4 ± 20.2	112 ± 38.2	0.700
Acute Phase Reactants	CRP high	Normal	
Serum copper level (µg/dL)	112.7 ± 23	158	0.154
Respiratory Function	Sufficient	Insufficient	
Serum copper level (µg/dL)	105.1 ± 23.3	120.1 ± 22.5	0.277
Dietary zinc intake (%DRI))	81.5 ± 20.8	112.5 ± 23.6	0.015
Pancreatic Function	Sufficient	Insufficient	
Serum copper level (µg/dL)	115.5 ± 29.7	112.3 ± 22.6	1.000

ERS: erythrocyte sedimentation rate; CRP: C reactive protein. %DRI: percentage of dietary reference intake.

**Table 3 nutrients-12-03344-t003:** Significant correlations and regression analysis between serum copper and nutritional parameters.

Nutritional Parameters	Serum Copper Level			
	Spearman’s Rho Test		Linear Regression Analysis	
	*r*	*p*-Value	*R* ^2^	*p*-Value
Body mass index	0.489 *	0.046	0.236	0.048
Average conduction velocity	0.517 *	0.040	0.275	0.037
Vitamin C intake	−0.651 **	0.040	0.270	0.039
Serum zinc	0.467	0.059	0.376	<0.0001
Cardiovascular risk index	0.51	0.045	0.26	0.045
Complement C3	0.616 *	0.014	0.350	0.020
Complement C4	0.477 **	0.001	0.535	0.002
Lymphocytes NK CD16+56	0.559 *	0.024	0.263	0.042
**Spearman’s Rho Test**	**Copper/Zinc Ratio**		**Zinc/Copper Ratio**	
	***r***	***p*-Value**	***r***	***p*-Value**
Zinc/copper ratio	0.998 **	0.000		
Protein intake (mg/d)	0.652 **	0.006	0.665 **	0.005
Monosaturated lipids intake (% DRI)	0.703 **	0.002	−0.691	0.003
Polyunsaturated lipids intake (% DRI)	−0.584 *	0.018	0.584	0.018
Niacin intake (% DRI)	0.641 **	0.007	−0.670 **	0.005
Calcium intake (% DRI)			0.507 *	0.045
Triglycerides	0.558 *	0.020	−0.542 *	0.025
Serum iron			0.493 *	0.045
Gamma glutamyl transpeptidase	0.574 *	0.016	−0.573 *	0.016
Monocytes	−0.532 *	0.034	0.541 *	0.031

Spearman correlation: ** correlation is significant at the 0.01 level (two-tailed), * correlation is significant at the 0.05 level (two-tailed).

**Table 4 nutrients-12-03344-t004:** Serum copper and iron levels, anemia biomarkers and number of lymphocytes (*N* = 17 *).

Age Years	Serum Copper µg/dL	Serum Iron µg/dL	Hemoglobin g/L	Mean Corpuscular Volume μg/m^3^	Lymphocytes Cell/mm^3^
2	113	49	13.2	82	6580
4	90	79	13.8	82.6	3400
6	158	46	13.5	79.7	3830
8	139	25	15	80.8	3710
8	108	84	14.2	88.9	3440
9	145	108	14.4	85.7	2490
9	121	79	14.35	87.8	3146
13	121	47	14.8	80.5	1900
15	69	135	14.3	89.4	1610
15	100	69	15.1	88.5	3970
16	105	89	15.6	86.3	4120
18	85	69	11.7	92	2110
20	113	129	15.4	84.4	3200
23	125	141	17.8	96.2	2490
23	100	45	13.4	97.8	2270
25	139	92	14.4	101.2	2920
31	91	59	13.1	89.3	2300

* 17 cystic fibrosis patients were screened, included and analyzed.

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
