# Peer review of "Copper and Copper/Zinc Ratio in a Series of Cystic Fibrosis Patients"

_nutrients, 2020, doi:10.3390/nu12113344_

Round 1

Reviewer 1 Report

Dear Authors

Your article concerning Copper/Zinc ratio in a series of cystic fibrosis patients is pleasant to read. I have a few remarks and questions.

Line 79-81: Sentence is not clear or incomplete

Line 131: A dietary survey of 72 hours isn’t it a bit short to support conclusions?

Increasing attention to the ratio Cu/ Zn is very interesting and worth of more studies.

Best regards

Author Response

Changes introduced in the manuscript nutrients-980682

We acknowledge the reviewers for their useful comments, which helped to improve the quality of the manuscript.

  • Comment 1. Lines 79-81: Sentence is not clear or incomplete

Author’s response: Lines 88-90 Thank you for your comment. The sentence was changed as follow: ‘Furthermore, copper deficiency can occur in premature infants who are fed formulas with inadequate copper content, newborns with chronic diarrhea or malnutrition, patients undergoing prolonged dialysis or severe burns.’

  • Comment 1. Line 131: A dietary survey of 72 hours isn’t it a bit short to support conclusions?

Author’s response: Lines 142-143. Dietary intake is difficult to measure, and any single method cannot assess dietary exposure perfectly. However, a prospective 3-day intake record is the most accepted method of knowing the intake that the patient does in clinical practice. A sentence was added as follow: ‘A 24-hour food diary was recorded during 3 days, and the collection of information on the habitual diet of the participants was verified by an in-depth interview (approximately 90 minutes).’

Thank you for your contribution, we really appreciate the review made that enriches the presentation of this research work.

Best regards

Reviewer 2 Report

This study represents a relevant take on the body copper status of cystic fibrosis (CF) patients. CF is associated with various nutrient deficiencies, thus studies on the micronutrient supply of CF patients are needed. By examining the association of copper serum levels with those measured for zinc, as well as with characteristic factors of the CF and with several nutritional parameters, this study highly contributes to the overall understanding of this disease. However, I still see a couple of problems with the manuscript as it is:

Major:

The manuscript needs editing of English language, including grammatical errors and some word choices, as this sometimes complicates the reading flow and understanding of the text (see also minor revisions):

For example:

Lines 47-49: ‘With increasing longevity, the burden and prevalence of comorbidities have  increased, not only CF-related diabetes (CFRD), CF-related liver disease (CFLD), CF-related kidney and bone disease, but also obesity and overweight that have appeared in this specific population’ are not easily understandable. I would suggest changing it to: ‘With increasing longevity, the burden and prevalence of comorbidities increase, accordingly CF-related diabetes (CFRD), CF-related liver disease (CFLD), CF-related kidney and bone disease along with increased chance for obesity, and overweight were reported in CF patients’.

Lines 202-203: ‘Cystic Fibrosis is a multisystem disorder involving pulmonary, gastrointestinal, endocrine, musculoskeletal, male genitourinary system, and sinuses’. I would suggest rephasing this sentence to: ‘involving the pulmonary,……endocine, musculoskeletal, and the male genitourinary system, as well as sinuses’

Line 17: To underline the relevance of you study it could be useful to state the need for a stable and sufficient micronutrient supply during as well as the general occurrence of micronutrient deficiencies during this disease early on in the abstract.

Line 25-26: Please rephrase the sentence to show that there was no significant correlation between the copper concentrations in serum and the respective parameter analyzed

Line 28-29: Please emphasize the importance of your findings. Why measuring Cu/Zn ratios when the same information could be obtained measuring serum Zn levels? What is the advantage or gain of this approach?

Line 30: Please extend the keywords with ‘serum copper/zinc ratio’.

Line 44: Please elaborate what kind of ‘penetrance’ you mean? Do you mean the severity and consequences of the diseases?

Line 49: Please specify which kind of population you refer to? Do you mean CF patients?

Line 51: Please elaborate which gene mutation you mean? Do you mean the gene mutation of the disease?

Line 164-165: I would suggest rephrasing this sentence to increase its understanding. For example: ‘There was no significant difference in serum copper comparing patients with high and low abdominal ultrasound, sufficient and insufficient respiratory or pancreatic function, as well as patients with or without respiratory colonization (Table 2).’

Line 174: Please elaborate what you define as a ‘normal’ serum Cu/Zn ratio?

Lines 176-181: I would suggest including information on the respective Table (I think it is Table 3) where the results that you present in this section are listed.

Line 182: Please refer to Table 1 in the text (materials and methods?) and transfer it to the respective section. I would also suggest transferring Figure 1 to the materials and methods section show Table 2 before Figure 2 and 3.

Lines 190-193: I would suggest including Figure 4 and 5 and their main results to the result part.

Line 215 and Lines 311-312: Please elaborate what you mean by ‘normal’? I think you refer to the range of copper or zinc serum levels, respectively, that are considered to represent an adequate copper supply.

Line 219-226: What is the take home message for this paragraph?

Line 256: Please include information on the bone densitometry analysis that you refer to in the text.

Lines 264-166: I would suggest transferring this part into the results section.

Lines 281-289: Would it be possible to provide the respective data (e.g. Cu levels, biomarkers for anemia, lymphocyte numbers) that you refer to into the Supplemental Material of this manuscript?

Line 318: Do you mean that serum copper and zinc levels did not correlate with each other and there was no association of their serum levels with the age of the patients?

Minor:

Line 18: Please specify: serum copper/zinc ratios

Line 34: Please change ‘White’ to ‘white’. As this is an adjective and written with small letters, even in a political context.

Line 37: changes ‘encrypts’ to ‘encode’

Line 50: linked directly to factors

Line 124: change ‘with’ to ‘of’

Line 111: change to pancreatic sufficiency (PS) and insufficiency (PI)

Line 166: significantly

Line 167: teenager

Line 227: Please change the term ‘gender’ to ‘sex’.

Line 229: ‘not’ significant

Line 238: --which is ‘in agreement’ with our results

Line 240: malnutrition

Line 241: higher BMI

Line 243: Not one

Line 290: the study of Dizdar et al.

Line 327: …might indicate that’..

Line 356: …serum copper significantly correlated with serum zinc…

Author Response

Changes introduced in the manuscript nutrients-980682

We acknowledge the reviewers for their useful comments, which helped to improve the quality of the manuscript.

  • Comment 1. Lines 47-49: ‘With increasing longevity, the burden and prevalence of comorbidities have  increased, not only CF-related diabetes (CFRD), CF-related liver disease (CFLD), CF-related kidney and bone disease, but also obesity and overweight that have appeared in this specific population’ are not easily understandable. I would suggest changing it to: ‘With increasing longevity, the burden and prevalence of comorbidities increase, accordingly CF-related diabetes (CFRD), CF-related liver disease (CFLD), CF-related kidney and bone disease along with increased chance for obesity, and overweight were reported in CF patients’.

Author’s response: Lines 53-56. Thank you for your comment the paragraph was modified according to your suggestion as follow: ‘With increasing longevity, the burden and prevalence of comorbidities increase, accordingly CF-related diabetes (CFRD), CF-related liver disease (CFLD), CF-related kidney and bone disease along with increased chance for obesity, and overweight were reported in CF patients.’

  • Comment 2. Lines 202-203: ‘Cystic Fibrosis is a multisystem disorder involving pulmonary, gastrointestinal, endocrine, musculoskeletal, male genitourinary system, and sinuses’. I would suggest rephasing this sentence to: ‘involving the pulmonary,……endocine, musculoskeletal, and the male genitourinary system, as well as sinuses’

Author’s response: Lines 231-232: Thank you for your comment. The paragraph was changed according to your suggestion as follow: ‘Cystic Fibrosis is a multisystem disorder involving pulmonary, gastrointestinal, endocrine, musculoskeletal, and the male genitourinary system, as well as sinuses.’

  • Comment 3. Line 17: To underline the relevance of you study it could be useful to state the need for a stable and sufficient micronutrient supply during as well as the general occurrence of micronutrient deficiencies during this disease early on in the abstract.

Author’s response: Line 17: Thank you for your comment. The paragraph was changed as follow: ‘Cystic Fibrosis (CF) patients, require a stable and sufficient supply of micronutrients.’

  • Comment 4. Lines 25-26: Please rephrase the sentence to show that there was no significant correlation between the copper concentrations in serum and the respective parameter analysed.

Author’s response: Lines 26-28: Thank you for your comment. The paragraph was changed as follow: ‘There was no significant correlation between serum copper concentrations and respiratory and pancreatic function, respiratory colonization, and abdominal ultrasound.’

  • Comment 5. Lines 28-29: Please emphasize the importance of your findings. Why measuring Cu/Zn ratios when the same information could be obtained measuring serum Zn levels? What is the advantage or gain of this approach?

Author’s response: Lines 32-33: Thank you for your suggestion. A sentence was added as follow: ‘The measurement of serum zinc alone does not show zinc status. However, the Cu/Zn ratio may be an indicator of zinc deficiency and the inflammatory status of CF patients.’

  • Comment 6. Line 30: Please extend the keywords with ‘serum copper/zinc ratio’.

Author’s response: Lines 34-35: The keywords was added.

  • Comment 7. Line 44: Please elaborate what kind of ‘penetrance’ you mean? Do you mean the severity and consequences of the diseases?

Author’s response: Line 49-51: Thank you for your suggestion. The sentence was changed as follow: ‘However, pancreatic disease presents the highest penetrance in severity and consequences of CF, as the pancreas is one of the first organs to be affected by this disease.’

  • Comment 8. Line 49: Please specify which kind of population you refer to? Do you mean CF patients?

Author’s response: Line 56: Thank you for your comment. The specific population is CF patients. This changed was done according to comment 1, too.

  • Comment 9. Line 51: Please elaborate which gene mutation you mean? Do you mean the gene mutation of the disease?

Author’s response: Lines 56-58: Thank you for your comment. Yes, we do. The changed was done as follow: ‘CF is closely related to poor nutritional status linked directly to factors associated with the genetic mutation underlying this disease.’

  • Comment 10. Lines 164-165: I would suggest rephrasing this sentence to increase its understanding. For example: ‘There was no significant difference in serum copper comparing patients with high and low abdominal ultrasound, sufficient and insufficient respiratory or pancreatic function, as well as patients with or without respiratory colonization (Table 2).’

Author’s response: Lines 177-180: Thank you for your suggestion. The sentence was changed as follow: ‘There was no significant difference in serum copper comparing patients with high and low abdominal ultrasound, sufficient and insufficient respiratory or pancreatic function, as well as patients with or without respiratory colonization (Table 2).'

  • Comment 11. Line 174: Please elaborate what you define as a ‘normal’ serum Cu/Zn ratio?

Author’s response: Line 190: Thank you for your comment. The sentence was changed as follow: ‘Mean Cu/Zn ratio of 1.32 ± 0.28 (Q1-3, 1.19-1.38) and range from 0.73 to 2.0, was higher than normal value with a range from 0.7 to 1.0 and…’

  • Comment 12. Lines 176-181: I would suggest including information on the respective Table (I think it is Table 3) where the results that you present in this section are listed.

Author’s response: Line 207: Thank you for your comment. The results appear in the Table 3 in results section.

  • Comment 13. Line 182: Please refer to Table 1 in the text (materials and methods?) and transfer it to the respective section. I would also suggest transferring Figure 1 to the materials and methods section show Table 2 before Figure 2 and 3.

Author’s response: Thank you for your suggestion. The reference of Table 1 (Line 207) is in results section, Figure 1 in materials and methods (Line 126) section and Table 2 (Line 212) appears in the results section before Figure 2 (Line 213) and 3 (Line 217). The position of respective tables and figures were changed in the manuscript.

  • Comment 14. Lines 190-193: I would suggest including Figure 4 and 5 and their main results to the result part

Author’s response: Line 197: Thank you for your suggestion. The reference of both figures appears in the results section.

  • Comment 15. Lines 215 and 354: Please elaborate what you mean by ‘normal’? I think you refer to the range of copper or zinc serum levels, respectively, that are considered to represent an adequate copper supply.

Author’s response: Lines 246 and 355-356: Thank you for your comment. They refer that mean were into the range of copper and zinc serum concentration considered adequate, respectively. The normal range from both micronutrients was added.

  • Comment 16. Line 219-226: What is the take home message for this paragraph?

Author’s response: Lines 250-257: The paragraph was changed for its better understanding as follow: ‘Although low levels of serum and plasma copper, ceruloplasmin, and superoxide dismutase in red blood cells can show a severe copper deficiency, they are not sensitive to a marginal copper state [47]. Furthermore, the response to inflammation and infection can alter serum copper and not determine its deficiency [48]. According to different studies, pediatric reference intervals for serum copper are often difficult to establish [49]. Moreover, patients with CF serum copper and ceruloplasmin levels show variable results [50]. We must bear in mind that the consequences of borderline copper deficiency may have little effect on a normal individual but may have more serious consequences for CF patients [48]. Therefore, its assessment is essential for CF patients.’

  • Comment 17. Line 256: Please include information on the bone densitometry analysis that you refer to in the text.

Author’s response: Lines 298-299: Thank you for your comment. A sentence was added as follow: ‘That is, bone densitometry by ultrasound was normal and no patient with CF was at risk of osteoporosis [33]’.

  • Comment 18. Lines 264-266: I would suggest transferring this part into the results section.

Author’s response: Lines 201-206: Thank you for your suggestion. This information was transferred into the results section as follow: ‘Mean basal EE was lower than theoretical (p = 0.001) but was adequate according to the World Health Organization (WHO)’s recommendation (p = 0.074). The mean diet was a hyper protein with an adequate carb, fiber and EI. The diet was adequate except for the low iodine intake. Serum copper did not correlate with zinc, calcium, magnesium, and iron intakes. CF patients with RI had more zinc intake (112.5%DRI) than RS (81.5%DRI, p = 0.015). Only vitamin C intake had a negative association with serum copper (Table 3).’

  • Comment 19. Lines 281-289: Would it be possible to provide the respective data (e.g. Cu levels, biomarkers for anemia, lymphocyte numbers) that you refer to into the Supplemental Material of this manuscript?

Author’s response: Lines 325-328. Thank you for your suggestion. The Table 4 was added as Supplemental Material.

  • Comment 20. Line 318: Do you mean that serum copper and zinc levels did not correlate with each other and there was no association of their serum levels with the age of the patients?

Author’s response: Lines 360-362: Thank you for your suggestion. Yes, I do. The sentences was changed as follow: ‘Although serum copper and zinc levels did not correlate with each other and there was no association of their serum levels with the age of the patients, linear regression analysis showed that serum copper had a significant association with serum zinc when adjusted for age.’

  • Comment 21. Line 18: Please specify: serum copper/zinc ratios.

Author’s response: Line 19: The word was added, and the phrase was changed.

  • Comment 22. Line 34: Please change ‘White’ to ‘white’. As this is an adjective and written with small letters, even in a political context.

Author’s response: Line 39: Thank you for your suggestion. The word was changed.

  • Comment 23. Line 37: changes ‘encrypts’ to ‘encode’

Author’s response: Line 42: Thank you for your comment. The word was changed.

  • Comment 24. Line 50: linked directly to factors

Author’s response: Line 57: Thank you for your suggestion. The word was changed.

  • Comment 25. Line 124: change ‘with’ to ‘of’

Author’s response: Line 135: Thank you for your comment. The word was changed

  • Comment 26. Line 111: change to pancreatic sufficiency (PS) and insufficiency (PI)

Author’s response: Line 122: Thank you for your suggestion. The position of this word was changed.

  • Comment 27. Line 166: significantly

Author’s response: Line 180: Thank you for your comment. The word was changed.

  • Comment 28. Line 167: teenager

Author’s response: Line 182: Thank you for your comment. The word was changed.

  • Comment 29. Line 267: Please change the term ‘gender’ to ‘sex’.

Author’s response: Thank you for your suggestion. The word was changed.

  • Comment 30. Line 268: ‘not’ significant

Author’s response: Thank you for your comment. The word was changed.

  • Comment 31. Line 278: --which is ‘in agreement’ with our results

Author’s response: Thank for your suggestion. The word was changed.

  • Comment 32. Line 281: malnutrition

Author’s response: Thank you for your comment. This word was changed.

  • Comment 33. Line 281: higher BMI

Author’s response: Thank you for your suggestion. This word was changed.

  • Comment 34. Line 243: Not one

Author’s response: Line 283: Thank you for your comment. This word was changed.

  • Comment 35. Line 290: the study of Dizdar et al.

Author’s response: Line 333: Thank you for your suggestion. This phrase was changed.

  • Comment 36. Line 327: …might indicate that’..

Author’s response: Lines 370-371: Thank you for your suggestion. This phrase was changed.

  • Comment 37. Line 356: …serum copper significantly correlated with serum zinc…

Author’s response: Line 400: Thank you for your comment. The phrase was changed.

Thank you for your contribution, we really appreciate the review made that enriches the presentation of this research work.
